# Jatoba (*Hymenaea courbaril* L.) Pod Residue: A Source of Phenolic Compounds as Valuable Biomolecules

**DOI:** 10.3390/plants13223207

**Published:** 2024-11-15

**Authors:** Gabriela Lucca Del Angelo, Isabela Silva de Oliveira, Bianca Rodrigues de Albuquerque, Samanta Shiraishi Kagueyama, Tamires Barlati Vieira da Silva, José Rivaldo dos Santos Filho, Maria Inês Dias, Carla Pereira, Ricardo C. Calhelha, Maria José Alves, Ariana Ferrari, Natalia Ueda Yamaguchi, Acácio Antonio Ferreira Zielinski, Adelar Bracht, Rosane Marina Peralta, Rúbia Carvalho Gomes Corrêa

**Affiliations:** 1Postgraduate Program in Clean Technologies, Cesumar University—UNICESUMAR, Maringa 87050-390, PR, Brazil; gabriela-lucca@hotmail.com (G.L.D.A.); isabelasilva_12@hotmail.com (I.S.d.O.); 2Centro de Investigação de Montanha (CIMO), Instituto Politécnico de Bragança, Campus de Santa Apolónia, 5300-253 Bragança, Portugal; bianca.albuquerque@ipb.pt (B.R.d.A.); maria.ines@ipb.pt (M.I.D.); carlap@ipb.pt (C.P.); calhelha@ipb.pt (R.C.C.); maria.alves@ipb.pt (M.J.A.); 3Laboratório Associado para a Sustentabilidade e Tecnologia em Regiões de Montanha (SusTEC), Instituto Politécnico de Bragança, Campus de Santa Apolónia, 5300-253 Bragança, Portugal; 4Department of Biochemistry, State University of Maringá, Maringa 87020-900, SP, Brazil; ra124361@uem.br (S.S.K.); tamiresbarlati93@gmail.com (T.B.V.d.S.); joserivaldosf01@gmail.com (J.R.d.S.F.); abracht@uem.br (A.B.); rmperalta@uem.br (R.M.P.); 5Postgraduate Program in Health Promotion, Cesumar University—UNICESUMAR, Maringa 87050-390, PR, Brazil; ariana.ferrari@unicesumar.edu.br; 6Cesumar Institute of Science, Technology and Innovation—ICETI, Maringa 87050-390, PR, Brazil; 7Department of Energy and Sustainability, Federal University of Santa Catarina, Araranguá 88905-120, SC, Brazil; natalia.ueda@ufsc.br; 8Department of Chemical Engineering and Food Engineering, Federal University of Santa Catarina, C.P. 476, Florianópolis 88040–900, SC, Brazil; acacio.zielinski@ufsc.br

**Keywords:** food waste recovery, phenolic compounds, antioxidant potential, antiproliferative potential, antimicrobial activity, upcycling

## Abstract

This study aimed at investigating the chemical composition and a selected group of bioactivities of jatoba (*Hymenaea courbaril* L.) pod residue. An aqueous extract (deionized water; AE) and a hydroethanolic extract (ethanol: deionized water, 70:30 *v*/*v*; ETOH) were obtained via maceration. Ten phenolic compounds were characterized via LC-DAD-ESI/MSn: seven procyanidins, two quercetin derivatives and one taxifolin derivative, with dimers and trimers of procyanidins being the main components of both extracts. Total phenolic compound levels of 2.42 ± 0.06 and 11 ± 1 mg/g were found in AE and ETOH, respectively; however, only seven compounds were identified in ETOH. The jatoba pod residue extracts showed notable antioxidant activities: ETOH had greater antioxidant potential in the OxHLIA and DPPH assays (IC_50_ = 25.4 μg/mL and 0.71 μg/mL, respectively); however, EA demonstrated greater potential in the FRAP system (IC_50_ = 2001.0 µM TE/mg). Only AE showed antiproliferative potential, being effective against cell lines of gastric (GI_50_ = 35 ± 1 µg/mL) and breast (GI_50_ = 89 ± 4 µg/mL) adenocarcinomas. Likewise, only AE showed modest anti-inflammatory potential (IC_50_ = 225 ± 2 µg/mL) in mouse macrophages. Bacteriostatic effects against bacteria were exerted by both extracts. *Enterococcus faecalis* and *Listeria monocytogenes* (MICs = 2.5 mg/mL) were especially sensitive to the ETOH extract. Taken together, the results suggest potential for jatoba pod residue as a source of molecules with biological activities and with possible industrial applications.

## 1. Introduction

Phenolic compounds, present in fruits and vegetables, are said to be beneficial to health. This is mainly due to their antioxidant, anti-inflammatory, antimicrobial and anticarcinogenic properties [1]. There is a widespread notion that intake of foods containing these compounds can be helpful in the prevention of several chronic diseases that are frequently associated with oxidative stress [2,3].

*Hymenaea courbaril* L., known as jatoba, is an arboreal legume of the Caesalpiniaceae family that occurs abundantly in Brazilian forests. It is found in diverse biomes such as the Cerrado and the Pantanal [4]. It has been widely used in the recovery of deforested areas and afforestation [5], and showed potential for phytoremediation [6]. The species has economic value for providing high quality wood, but also because its leaves, bark, sap, resin and fruits contain significant levels of bioactive phytochemicals. The latter justify its use in traditional medicine as a raw material for obtaining incense, cosmetics and food ingredients, as well as natural tonics, fortifiers and energizers [7,8] (Figure 1). In addition to phenolic constituents such as procyanidins and catechins [9,10], several other compounds—mainly diterpenes and sesquiterpenes of the enantio-labdanoid and enantio-halimane types—have been isolated from seeds, leaves and trunk bark of *H. courbaril* [11,12,13]. Chemical screening of the edible fruit-derived pulp, a yellowish powder with a sweet taste, revealed the presence of linolenic acid as well as sucrose [7]. The variety of effects on biological systems that were described for the various preparations derived from *H. courbaril* seem to be caused by a multivariate chemical composition. Among the reported biological activities, there are antioxidant [8], anti-inflammatory [14], antiviral and antiproliferative [15] properties, as well as antimicrobial [16,17] and larvicidal [18,19] effects.

New and highly-valued molecules are currently being isolated from agricultural by-products and used for the development of foods or food ingredients with the purpose of conserving and/or adding functionality, thus promoting efficiency in the use of resources and circularity [20]. On this respect, the residues from the commercial exploitation of the jatoba tree remain underexplored. Despite the several above-mentioned studies on the different parts of the *H. courbaril* plant, there are relatively few communications on the jatoba pod shell, which is a residue arising from the industrial processing of the pulp and is usually discarded [19,21].

The purpose of the present investigation was, thus, to rate the phenolic profile of the *Hymenaea courbaril* pod shell extracts (aqueous and hydroethanolic) and to quantify possible characteristic biological actions.

## 2. Material and Methods

### 2.1. Plant Material

The mature pods of *Hymenaea courbaril* were collected at the Iguatemi Experimental Farm of the State University of Maringá, district of Iguatemi (23°25′ S; 51°57′ W; 550-m altitude), Parana State, Brazil. After identification, a voucher specimen was deposited under the code HUEM000020252 in the Herbarium of the State University of Maringá (*Herbanário da Universidade Estadual de Maringá*—HUEM).

The jatoba fruits were cleaned with water and dried at room temperature. Separation of the yellow floury pulp, containing the seeds, from the hard shells of each pod was performed manually. After separation, the residue pods were broken into smaller pieces with the aid of a hammer and ground into a fine powder. The latter was stored away from light and moisture and placed under refrigeration.

### 2.2. Extraction Procedure

A preliminary survey was carried out using five distinct extracting solutions. The antioxidant activity of each extract was determined using the iron reducing power (FRAP) assay. The efficiency in extracting antioxidant activity obeyed the following descending sequence: deionized water > ethanol/water mixture (70:30, *v*/*v*) > methanol/water mixture (80:20, *v*/*v*) > acetone/water mixture (80:20, *v*/*v*) > ethyl acetate. Also taking into account their green and GRAS features, the first two were chosen to be used in this work.

The extracts, aqueous (AE) and hydroethanolic (ETOH), reflect a proportion of pod residue/extractor solution of 1:20, i.e., 20 mL of extractor medium was added for 1 g of residue. The sealed vials were shaken for 2 h at 130 rpm at room temperature (25 °C) and kept in the dark. This procedure was repeated three times. The combined extracts were centrifuged at 1792× *g* for 15 min. The supernatant derived from the hydroethanolic extraction was evaporated at 35 °C as a means of removing ethanol. Lyophilization was the last step before storing the material at −20 °C until the analytical procedures.

### 2.3. Phenolic Compounds

Before analysis, the jatoba pod shell extracts were reconstituted to a concentration of 10 mg/mL using an 80:20 (*v*/*v*) methanol/water solution. Chromatography was conducted in a UHPLC system. This system is equipped with a diode array detector and connected to an electrospray ionization mass spectrometer. Details about the chromatographic separation (column, solvents, wavelength of detection) and mass spectrometric detection are described elsewhere [22].

Identification of the compounds relied on: (a) retention time comparisons with standards; (b) analysis and comparison of UV-visible and mass spectra with those standards; (c) literature data. For quantification, calibration curves (absorbance of UV light versus concentration) constructed with available standards were used. If commercial standards were not available, quantification was performed using the most closely related standard. Contents were given as milligrams per gram extract.

### 2.4. Antioxidant Activity

The OxHLIA assay was performed using the experimental protocol described previously, as well as the same procedure for calculating the corresponding IC_50_ values [23]. The latter corresponds to the extract concentration that protects half of the erythrocytes from the hemolytic effects of 2,2′-azobis (2-methylproprionamide) dihydrochloride when the measurements are performed over a period of 60 min.

Inhibition of the generation of thiobarbituric acid reactive substances (TBARS) and the DPPH radical scavenging activity were assessed following a previously described methodology [24]. The final outcomes were presented as IC_50_ values. The FRAP assay was conducted as previously described [25]. Standard curves were constructed using Trolox (R^2^ = 0.999), and the results were expressed as micromol equivalents of Trolox (TE) per milligram of lyophilized extract.

### 2.5. Cytotoxic Potential

Cytotoxicity assays were conducted with four human tumor cell lines: gastric adenocarcinoma (AGS), colorectal adenocarcinoma (CaCo2), breast adenocarcinoma (MCF-7) and lung carcinoma (NCI-H460). Cultivation and handling of the cells were described previously, as well as the sulforhodamine B assay that was used [13]. Results were given as the concentration causing 50% inhibition of cell proliferation (GI_50_). The commercial drug ellipticine was utilized as a positive control.

### 2.6. Anti-Inflammatory Activity

The anti-inflammatory potential was evaluated using the inhibition of the lipopolysaccharide (LPS)-induced nitric oxide (NO) production in mouse macrophages RAW 264.7 as an indicator, as described previously [24]. The positive control was dexamethasone (50 μM) and the negative control was inferred from measurements carried out in samples without the addition of LPS, which allow for the observation of possible effects on the basal levels of NO. The results were expressed as extract concentrations (μg/mL) capable of producing 50% of inhibition of NO production (IC_50_).

### 2.7. Antimicrobial Potential

The antibacterial efficacies of the pod residue extracts were determined using the colorimetric method described previously [26]. The indicator parameter, MIC, was defined as the lowest concentration capable of inhibiting visible bacterial growth. Additionally, the minimum bactericidal concentration (MBC) was assessed by identifying the lowest concentration that completely prevented growth, thereby defining the MBC as the minimum concentration required to achieve bacterial death. Ampicillin (Gram-negative bacteria) and imipenem (Gram-positive bacteria) were the positive controls.

### 2.8. Determination of the Inhibitory Activity of Jatoba Bark Extracts on Elastase and Collagenase

The elastase inhibitory activity was assessed using a slightly modified version of an originally described procedure [27], employing N-succinyl-Ala-Ala-Ala-p-nitroanilide as the substrate. The release of p-nitroaniline was monitored spectro-photometrically at 405 nm. In the reaction mixture, the final concentrations of substrate and elastase were 0.4 mM and 0.00225 U/mL, respectively. The IC_50_ values were determined from assays that were run with various concentrations in the range of up to 100 µg/mL, with distilled water serving as the negative control. Epigallocatechin gallate (EGCG) was the positive control. All experiments were conducted in triplicate. The percentage of elastase inhibition was computed as follows:$$\mathrm{Efficiency}\;(\%)=\left(1-\frac{\mathrm{absorbance}\;\mathrm{of}\;\mathrm{the}\;\mathrm{sample}}{\mathrm{absorbance}\;\mathrm{of}\;\mathrm{the}\;\mathrm{negative}\;\mathrm{control}}\right)\times 100$$

The anti-collagenase activity was evaluated using a method predicated on the interaction between bacterial collagenase sourced from *Clostridium histolyticum* and the substrate N-[3-(2-furyl) acryloyl]-Leu-Gly-Pro-Ala, using the incubation medium and measurement procedures described elsewhere [28]. Absorbance was measured at 335 nm. To determine the IC_50_ values, sample solutions were prepared to achieve final concentrations of up to 800 µg/mL, with distilled water serving as a negative control. Epigallocatechin gallate (EGCG) was the positive control. The percentage of collagenase inhibition was computed using the same formula mentioned above.

### 2.9. Statistical Analysis

All results were expressed as mean values plus standard deviations (SDs) of three separate experiments. Analysis of variance (ANOVA) or the *t* test were applied, according to the context. ANOVA was followed by Tukey’s *post hoc* test with a significance level of 5% (*p* ≤ 0.05).

## 3. Results and Discussion

### 3.1. Phenolic Profile of the Jatoba Pod Residue Extracts

The concentrations of the phenolic compounds found in the jatoba pod residue extracts are presented in Table 1.

Ten phenolic compounds were tentatively identified: seven procyanidins, two quercetin derivatives and one taxifolin derivative. Peaks 1, 2 and 4 showed the same molecular ion ([M-H]^−^ at *m*/*z* 577), releasing MS^2^ fragments at *m*/*z* 451 (−126 u), 425 (−152 u) and 407 (−152 u). −18 u). This fragmentation behavior could be explained by heterocyclic ring fission (HRF) and retro-Diels-Alder fragmentation (RDA). Also, two other fragment ions at *m*/*z* 287 and 286, corresponding to the lower and upper (epi)catechin, were disassociated in the MS^2^. According to the literature, these peaks were identified as Type B (epi)catechin dimers [29].

Peaks 3, 5 and 6, with the same pseudo molecular ion [M-H]^−^ at *m*/*z* 865, and Peak 7 ([M-H]^−^ at *m*/*z* 1153), showed the same above-mentioned fragmentation pattern, being tentatively identified as Type-B (epi)catechin trimers and tetramer, respectively [29]. The presence of procyanidin dimers, trimers and tetramers in extracts of trunk bark [9] and sap from jatoba [11] have already been detected before.

Peak 8 ([M-H]^−^ at *m*/*z* 449) showed two MS^2^ fragments at *m*/*z* 303 and 285, perhaps from the loss of rhamnose and water moieties, which revealed a dihydroquercetin compound. This compound has been identified before as taxifolin-*O*-rhamnoside [9]. Thus, considering the mass spectral characteristics and literature data, this peak was provisionally identified.

Peak 9 ([M-H]^−^ at *m*/*z* 463), with a MS^2^ fragment at *m*/*z* 301, was identified as quercetin-3-O-glucoside based on its mass spectral characteristics and comparison with a commercial standard. Peak 10 ([M-H]^−^ at *m*/*z* 447) also released a fragment at *m*/*z* 301, corresponding to a quercetin molecule after the loss of a rhamnose unit (−146 u). This compound was identified in jatoba-do-cerrado (*Hymenaea stignocarpa* Mart.) extracts as quercetin-*O*-rhamnoside [21].

Compounds 2 and 4, both type B (epi)catechin dimers, were the major constituents of the pod residue samples, making up more than 43% of the phenolic content of both extracts (Table 1). Although the hydroethanolic extract showed higher levels for all phenolic compounds determined in common with the aqueous extract, compounds 1, 7 and 10 were detected only in the latter. The total phenolic content (TPC) determined for the hydroethanolic extract of jatoba pod residue (11 mg/g) was 4.5-fold higher than the aqueous extract (2.42 mg/g). The recovery of phenolic compounds from plant matrices is strongly influenced by solvent polarity, as polar solvents preferentially extract polar molecules. Consequently, hydroethanolic solutions serve as effective solvents for isolating phenolic compounds, combining compatibility with food-grade standards and an optimal polarity range to solubilize these bioactive compounds [30].

TPCs, based on milligrams of gallic acid equivalent (GAE) per gram of pulp from three jatoba samples, were estimated in a previous report as averaging 1.05 mg GAE/g of the *H. courbaril* pulp [31]. In a study in which the phenolics extraction from the jatoba pod shell was optimized [21], TPC values ranging from 16.289 to 32.424 mg GAE/g (dry matter) were found. More recently, a TPC of 490.533 mg TAE (tannic acid equivalent)/g of essential oil was obtained from the jatoba pod shell [19]. These values are possibly overestimated as the Folin Ciocalteu reagent, used in the assays of total phenolics, is not specific and reacts with many substances such as reducing sugars and ascorbic acid [32].

### 3.2. Biological Properties of the Jatoba Pod Residue Extracts

Four distinct antioxidant assays were used to evaluate the antioxidant potential of jatoba pod shell extracts (Table 2). The use of more than two methods is advisable because each of them has a particular reaction target [33]. This particularity frequently leads to different degrees of antioxidant activity in the various tests [32]. In this sense, complementary information on the antioxidant capacity of the investigated extracts/compounds were obtained. To our knowledge, this is the first time that such a set of methods has been employed to quantify the antioxidant capacity of *H. courbaril* samples.

The hydroalcoholic extract had higher antioxidant activity than the aqueous extract in the DPPH and OxHLIA assays, with IC_50_ values of 0.71 and 25.4 μg/mL, respectively (Table 2). These values are similar to those measured for the positive control, Trolox. In the FRAP assay, however, the aqueous extract showed the highest activity. Interestingly, in the TBARS system, the IC_50_ values recorded for our samples were almost 1000-fold higher than those for Trolox. The low protection efficiency of the pod residue extracts in the TBARS assay suggests that they would not be ideal for application as preservatives in lipid systems.

A bioassay-guided fractionation of an ethanolic extract of the stem bark of *H. Courbaril* was reported [14]. The crude extract, the ethyl acetate fraction and the methanolic fraction showed IC_50_ values of 30,770.18, 50,571.5 and 51,270.73 μg/mL, respectively, in the DPPH assay, whereas the positive control (Trolox) had an IC_50_ value of 2670.23 μg/mL. Using the same method, an evaluation of the antioxidant activity of the aqueous, ethanolic and hydroacetonic extracts from the pulp of jatoba-do-cerrado (*Hymenaea stigonocarpa* Hayne.) was conducted [34]. The hydroacetonic extract showed the highest antioxidant capacity among their samples (IC_50_ = 26.76 mg/mg), followed by the aqueous (IC_50_ = 117.07 mg/mg) and ethanolic (IC_50_ = 618.06 mg/mg) extracts.

More recently, the antioxidant capacities of pulp flour and fibrous pulp residue samples from *H. courbaril,* using DPPH and FRAP methods, were assessed [8]. For the jatoba pulp, IC_50_ values of 0.045 and 0.036 µM TE/mg, respectively, were found in the DPPH and FRAP systems. The corresponding pulp residue showed superior antioxidant capacities, with values of 0.072 and 0.051 µM TE/mg, respectively.

The antioxidant potential determined for the jatoba pod residue extracts may be related, inter alia, to the presence of procyanidins (Table 2). It was indeed demonstrated that methanolic extracts of açai seeds (*Euterpe oleracea* Mart.), rich in type A and B procyanidins, have high antioxidant capacity, confirmed by DPPH, TEAC and ORAC methods [35]. The antioxidant and antibacterial potentials of these extracts was attributed to their phenolic composition [35]. Likewise, grape seed proanthocyanidins (GSPs), molecules recovered from by-products of the grape juice and winery industries, hold a variety of potent pharmacological effects, including oxidative stress inhibition and anti-inflammatory and antitumor activities, all of which have been verified in in vitro and in vivo models, as well as clinical trials [36]. Furthermore, a recent study showed that different doses of proanthocyanidins (PCs) can regulate plasma lipid levels. This effect on circulating lipids deserves to be examined more closely due to its potential in preventing lipid metabolism disorders [37].

It has been reported that the biological activities of procyanidins are associated with their chemical structure and their degree of polymerization and galloylation [38]. Their antioxidant potential depends on the presence of the catechol unit on the aromatic B-ring, which acts as a donor of hydrogen atoms, stabilizing the free radicals. The ability to chelate metals and proteins, on the other hand, can be attributed to o-dihydroxy phenolic groups in their high molecular weight structure [39].

Table 3 summarizes the results of the evaluation of cytotoxic and anti-inflammatory potentials of the jatoba pod residue extracts. Only the aqueous extract (AE) showed antiproliferative potential, in addition to slight anti-inflammatory activity (IC_50_ = 225 ± 2 µg/mL). AE exhibited pronounced cytotoxic activities against gastric adenocarcinoma (GI_50_ = 35 ± 1 µg/mL) and breast adenocarcinoma (GI_50_ = 89 ± 4 µg/mL) cells, and less pronounced growth-inhibiting effects against colorectal and lung carcinoma cell lines.

It has been reported that a hydroethanolic seed extract of jatoba exhibits dose- and time-dependent cytotoxic action against the B16F10 melanoma cell line [40]. In addition, the same extracts were demonstrated to exert antigenotoxic effects on mice bone marrow cells, as well as antioxidant activity [40]. These effects could be due to the presence of flavones in the extract. Caryophyllene oxide was identified as the major bioactive compound in the leaf hexane extract of *H. courbaril*, which showed promising antiproliferative effects against androgen-independent PC-3 prostate cancer cells [41]. It is possible that the molecules involved in the antiproliferative activities of our AE extract are different from the phenolic compounds identified here (Table 1).

The antimicrobial potential of the jatoba pod residue extracts against clinical isolates, including bacteria considered to be food-borne pathogens, is expressed in terms of minimum inhibitory concentrations (MICs) and minimum bactericidal concentrations (MBCs) of the extracts, and is shown in Table 4.

The MIC values reveal that both extracts exhibited bacteriostatic effects against all of the tested bacteria, suggesting the presence of a broad spectrum of phytochemicals with antibiotic activity. The hydroethanolic extract demonstrated higher inhibitory potential than the aqueous extract, being especially effective against *Morganella morganii*, *Enterococcus faecalis*, *Listeria monocytogenes* and methicillin-resistant *Staphylococcus aureus* (MIC values = 2.5–5 mg/mL).

More expressive inhibitory effects than ours have been reported before when the in vitro antimicrobial potential of ethanolic extracts obtained from the trunk barks of *Hymenaea courbaril* and *Stryphnodendron adstringens* against clinical bacterial isolates were evaluated [16]. The combination of extracts showed potent synergistic antimicrobial activity, with MIC values of 0.03 mg/mL against *Acinetobacter baumannii*, *Escherichia coli* and *Staphylococcus aureus*. The authors of this study [16] detected alkaloids, coumarins, flavonoids, steroids/triterpenoids and tannins in their ethanolic extract of *H. courbaril*. In another study, the inhibitory effect of the essential oil extracted from the jatoba pod residue was evaluated [42]. Pronounced activity against the Gram-positive *S. aureus*, combined with synergistic modulatory effects upon association with antibiotics for clinical use, was observed.

The antibacterial activities of the jatoba pod residue extracts studied may be related, inter alia, to their procyanidin content (Table 3). It has been found, for example, that two procyanidins isolated from a laurel wood extract, namely cinnamtannin B1 and procyanidin B2, promoted inhibition of bacterial growth at high concentrations and prevention of biofilm formation at lower ones [43]. The authors of this study suggest that this natural extract may be promising in the market of natural food preservatives and/or natural disinfectants for processing equipment where foodborne pathogens are frequently found.

The jatoba pod residue extracts showed MIC values greater than 1.6 for all bacteria tested (Table 4), which some authors might classify as a weak inhibitor profile [32]. However, it should be considered that the bacteria used here are multidrug-resistant strains (clinical isolates) that, in turn, hold antibiotic resistance profiles substantially superior to those of the ATCC bacterial strain [44]. In other words, the results reported can be interpreted as an indicator of relevant antibacterial potential.

The concentration curves of the extracts versus % inhibition of elastase and collagenase were performed, and the numerical data were used to calculate the IC_50_ (Table 5). For comparison purposes, the same curves were generated using epigallocatechin gallate (EGCG), a known inhibitor of elastase and collagenase. The IC_50_ values for elastase inhibition by jatoba extracts were lower than those obtained with EGCG. The IC_50_ values for collagenase inhibition by jatoba extracts were higher than those obtained with EGCG, but within the same order of magnitude. The tensile strength and elasticity of the skin are derived from collagen and elastin, proteins that are highly flexible and dynamic. Consequently, their decomposition leads to visible skin damage, wrinkle formation and a loss of elasticity and resilience [45,46]. Natural-origin actives with anti-collagenase and anti-elastase activity stand out in the production of anti-aging cosmetics. Plant extracts contain polyphenols such as flavonoids, phenolic compounds and proanthocyanidins which react with ROS, neutralizing free radicals as part of plants’ secondary defense metabolism [47,48]. This process reduces the expression of proteinases harmful to the structural proteins of the extracellular matrix, thus allowing the skin to maintain its youthful appearance.

## 4. Conclusions

The present study represents certainly one of the earliest comprehensive analyses of the phenolic makeup and biological activities of the *Hymenaea courbaril* pod husk, a byproduct typically discarded by the fruit pulp industry. Our results not only confirm the economic and bioactive potentials of the whole *H. courbaril* tree, but also endorse the potential of its bio-residues, particularly the pod husks, as interesting repositories of bioactive compounds for industrial purposes. The significant antioxidant potentials of the extracts, as evidenced using various assays, along with their promising bacteriostatic effects, suggest that they may be used as effective natural preservative ingredients. This is true, in principle, for both food and cosmetic formulations but other applications should not be excluded. And last, but not least, their inhibitory effects on collagenase and elastase deserve further investigations for being utilized in the development of cosmetics in the area of skin tensile strength and elasticity preservation. These results support the sustainable utilization of this biomass, in line with the principles of circular bioeconomy, potentially reducing waste and contributing to more efficient resource use.

## Figures and Tables

**Figure 1 plants-13-03207-f001:**
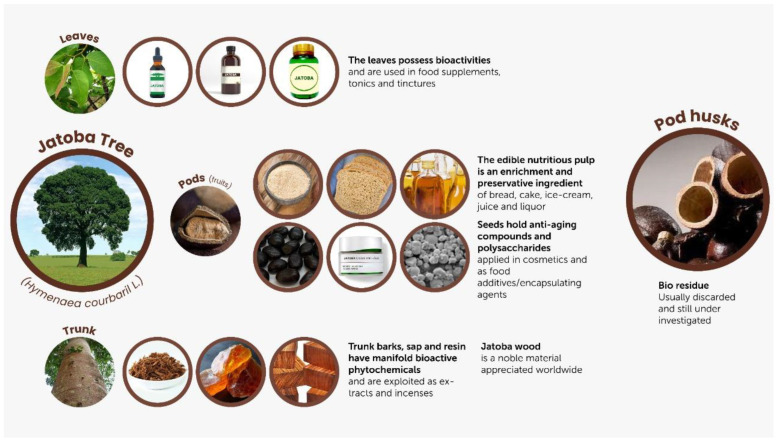
Most notable products and co-products of the jatoba tree (*Hymenaea courbaril* L.).

**Table 1 plants-13-03207-t001:** Retention times (R_t_), visible region maximal absorption wavelengths (λ_max_), mass spectrometry data, compound identification attempts, bibliographic references used to assist in the identification and quantification of the phenolic compounds found in the pod residue extracts (mean ± SD).

Peak	R_t_ (min)	λ_max_ (nm)	[M-H]^−^ (*m*/*z*)	Main Fragments MS^2^ (*m*/*z*)	Tentative Identification	Ref.	Quantification (mg/g of Extract)		*p* Value
							AE	ETOH	
1	4.86	280	577	451 (32), 425 (100), 407 (33), 289 (15), 287 (5)	Type-B (epi)catechin dimer	DAD-MSn [9,29]	0.101 ± 0.005	nd	-
2	5.09	280	577	451 (34), 425 (100), 407 (29), 289 (12), 287 (5)	Type-B (epi)catechin dimer	DAD-MSn [9,29]	0.745 ± 0.005	2.5 ± 0.2	0.007
3	6.17	280	865	739 (81), 695 (100), 577 (75), 575 (50), 425 (12), 289 (10), 287 (10)	Type-B (epi)catechin trimer	DAD-MSn [9,29]	0.15 ± 0.01	2.3 ± 0.4	0.013
4	6.95	280	577	451 (25), 425 (100), 407 (22), 289 (17), 287 (5)	Type-B (epi)catechin dimer	DAD-MSn [9,29]	0.309 ± 0.006	2.3 ± 0.5	0.001
5	9.83	280	865	739 (82), 695 (100),577 (76), 575 (53), 425 (10), 289 (10), 287 (10)	Type-B (epi)catechin trimer	DAD-MSn [9,29]	0.31 ± 0.01	1.27 ± 0.08	0.003
6	10.49	280	865	739 (79), 695 (100), 577 (81), 575 (50), 425 (15), 289 (10), 287 (10)	Type-B (epi)catechin trimer	DAD-MSn [9,29]	0.172 ± 0.005	1.45 ± 0.07	0.004
7	11.11	280	1153	865 (29), 713 (5), 577 (20), 575 (22), 289 (50)	Type-B (epi)catechin tetramer	DAD-MSn [9,29]	0.37 ± 0.02	nd	-
8	17.55	331	449	303 (100), 285 (50)	Taxifolin-*O*-rhamoside	DAD-MSn [9]	0.0845 ± 0.001	0.442 ± 0.002	<0.001
9	17.91	332	463	301 (100)	Quercetin-3-*O*-glucoside	DAD-MSn	0.0887 ± 0.0004	0.439 ± 0.001	<0.001
10	21.34	335	447	301 (100)	Quercetin-*O*-rhamnoside	DAD-MSn [21]	0.0886 ± 0.0005	nd	-
TPC							2.42 ± 0.06	11 ± 1	0.008

Results are means ± standard deviations which were compared using ANOVA. *p*-value < 0.05 means significant differences. Ref.—references for tentative identification; nd—not detected; TPC—Total phenolic compounds; Calibration curves used for quantification: catechin: y = 84,950x − 23,200; quercetin-3-*O*-glucoside: y = 34,843x − 160,173.

**Table 2 plants-13-03207-t002:** Antioxidant activities of the jatoba pod residue extracts (mean ± SD).

Antioxidant Activity	AE	ETOH	*p* Value	Trolox
TBARS (IC_50_ = μg/mL)	4659 ± 68	5367 ± 208	0.002	5.8 ± 0.6
OxHLIA (ΔT_60min_) (IC_50_ = μg/mL)	31.0 ± 0.4	25.4 ± 0.6	<0.001	21.8 ± 0.2
DPPH (IC_50_ = µg/mL)	5.40 ± 0.73	0.71 ± 0.33	0.003	0.11 ± 0.67
FRAP (µM TE/mg)	2001.0 ± 23.7	1378.9 ± 11.5	<0.001	-

Data are expressed as mean ± standard deviation. Statistical significance was evaluated through analysis of variance (ANOVA), with *p*-values < 0.05 for each row. IC_50_: concentration of the sample required to achieve 50% antioxidant activity. TE: Trolox equivalents, where Trolox is a synthetic antioxidant serving as a positive reference. The results are presented as mean ± standard deviation and analyzed by means of analysis of variance (ANOVA) and in each line *p*-value < 0.05. IC_50_ values: sample concentration providing 50% of the antioxidant activity. TE: Trolox equivalents. Trolox: synthetic antioxidant used as a positive control.

**Table 3 plants-13-03207-t003:** Cytotoxic and anti-inflammatory properties of the jatoba pod residue extracts.

	AE	ETOH	Ellipticine (µg/mL)
**Antiproliferative activity (GI_50_, µg/mL) ^1^**			
AGS (gastric adenocarcinoma)	35 ± 1	>400	1.23 ± 0.03
CaCo2 (colorectal adenocarcinoma)	173 ± 3	>400	1.21 ± 0.02
MCF-7 (breast adenocarcinoma)	89 ± 4	>400	1.02 ± 0.02
NCI-H460 (lung carcinoma)	186 ± 13	>400	1.01 ± 0.01
Vero (green monkey kidney cell line)	120 ± 7	>400	1.41 ± 0.06
	AE	ETOH	Dexametasone (µg/mL)
**Anti-inflammatory activity (IC_50_, µg/mL) ^2^**			
RAW 264.7 (mouse macrophages)	225 ± 2	>400	6.3 ± 0.4

^1^ GI_50_ values represent the concentration of the sample required to achieve 50% inhibition of cell growth in human tumor cell lines; ^2^ IC_50_ values indicate the concentration needed to inhibit 50% of nitric oxide (NO) production.

**Table 4 plants-13-03207-t004:** Antibacterial potential of *Hymenaea courbaril* pod residue extracts.

	ETOH		EA		Ampicillin (20 mg/mL)		Imipenem (1 mg/mL)		Vancomycin (1 mg/mL)	
	MIC	MBC	MIC	MBC	MIC	MBC	MIC	MBC	MIC	MBC
**Gram-negative bacteria**										
*Escherichia coli*	5	>20	5	>20	<0.15	<0.15	<0.0078	<0.0078	n.t.	n.t.
*Klebsiella pneumoniae*	>20	>20	>20	>20	10	20	<0.0078	<0.0078	n.t.	n.t.
*Morganella morganii*	5	>20	10	>20	20	>20	<0.0078	<0.0078	n.t.	n.t.
*Proteus mirabilis*	10	>20	10	>20	<015	<0.15	<0.0078	<0.0078	n.t.	n.t.
*Pseudomonas aeruginosa*	>20	>20	>20	>20	>20	>20	0.5	1	n.t.	n.t.
**Gram-positive bacteria**										
*Enterococcus faecalis*	2.5	>20	5	>20	<0.15	<0.15	n.t.	n.t.	<0.0078	<0.0078
*Listeria monocytogenes*	2.5	>20	5	>20	<0.15	<0.15	<0.0078	<0.0078	n.t.	n.t.
MRSA	5	>20	5	>20	<0.15	<0.15	n.t.	n.t.	0.25	0.5

MIC—minimum inhibitory concentration; MBC—minimum bactericidal concentration; n.t.—not tested; MRSA—Methicillin-resistant *Staphylococcus aureus*.

**Table 5 plants-13-03207-t005:** IC_50_ values of inhibition of elastase and collagenase by jatoba pod residue extracts.

Enzyme Inhibition	AE	ETOH	EGCG
Anti-elastase (IC_50_ = μg/mL)	17.48 ± 3.62 ^a^	19.57 ± 2.94 ^a^	36.71 ± 4.13 ^c^
Anti-collagenase (IC_50_ = μg/mL)	195.38 ± 17.20 ^a^	249.10 ± 22.50 ^b^	61.80 ± 12.52 ^c^

Pairs of values labeled with the same superscripts do not differ statistically from each other.

## Data Availability

The original contributions presented in the study are included in the article, further inquiries can be directed to the corresponding author/s.

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
