# Peer review of "Jatoba (Hymenaea courbaril L.) Pod Residue: A Source of Phenolic Compounds as Valuable Biomolecules"

_plants, 2024, doi:10.3390/plants13223207_

Round 1
Reviewer 1 Report
Comments and Suggestions for Authors
The by-products of agricultural processing are nutrient-rich. Discarding them causes resource waste and environmental pollution. Therefore, their comprehensive utilization has attracted significant attention. This manuscript systematically investigates the composition of phenolic compounds and their antioxidant and inhibitory properties in the residue of Jatoba pods. A total of ten phenolic compounds were identified in the extracts obtained through two different solvent extraction methods. The extracts exhibited strong antioxidant, anti-inflammatory, antiviral, and antibacterial activities. This manuscript demonstrates that Jatoba pod residues are a source of bioactive molecules, laying the groundwork for the high-value utilization of Jatoba pod residues. Overall, the manuscript is well-constructed, the research content is comprehensive, the analysis and discussion are appropriate, and the conclusions are valid. Here are some suggestions:
1. The overall similarity rate of the manuscript is particularly high, reaching 48%. The authors need to reduce it.
2. What is the basis for the extraction conditions of the two solvent extraction methods? Are there any references?
3. The discussion on the specific application prospects of Jatoba pod residues needs to be strengthened in the manuscript.
4. The reference format needs careful modification, such as italicizing species names.
Author Response
Comments 1: The overall similarity rate of the manuscript is particularly high, reaching 48%. The authors need to reduce it.
Answer 1: The requested alteration was done throughout the manuscript. All sections of the manuscript identified as containing similarities with other papers from our group were completely rewritten and are highlighted in yellow.
Comments 2: What is the basis for the extraction conditions of the two solvent extraction methods? Are there any references?
Answer 2: Thank you for the pertinent observation. A paragraph reporting the extraction study performed to select the extractor solutions used in our work was added to the '2.2 Extraction procedure' section (lines 106-112).
Comment 3: The discussion on the specific application prospects of Jatoba pod residues needs to be strengthened in the manuscript.
Answer 3: Thank you for the careful review of our work. A section dedicated to the specific application prospects of the investigated extracts has been added to the Conclusion section (lines 462-467).
Comments 4: The reference format needs careful modification, such as italicizing species names.
Answer 4: The references have been duly reviewed. All changes made are highlighted with yellow marker.
Reviewer 2 Report
Comments and Suggestions for Authors
In this work, authors investigated the chemical profile and biological properties of the Jatoba (Hymenaea courbaril L.) pod residue.
Please, find my suggestions:
- Please, correct any typos (es. "2.1. Plant material" all in italic!)
- Paragraph 2.1. Plant material: please, insert botanical identification and voucher specimen!
- a green monkey kidney cell 166 line (Vero) a non-tumor line was used. I suggest to use human non cancer cell line as control.
- Please, improve conclusion section.
Author Response
Comments 1: Please, correct any typos (es. "2.1. Plant material" all in italic!)
Answer 1: The requested alteration was performed. The text was carefully revised, which included language editing by a qualified professional. Both rewritten parts and insertions suggested by the reviewers are highlighted in yellow in the manuscript.
Comments 2: Paragraph 2.1. Plant material: please, insert botanical identification and voucher specimen!
Answer 2: Thank you for the pertinent observation. A paragraph reporting the botanical identification was added to the ‘2.1 Plant material’ section (lines 92-96).
Comments 3: A green monkey kidney cell 166 line (Vero) a non-tumor line was used. I suggest to use human non cancer cell line as control.
Answer 3: Using Vero cells as a control in cytotoxicity assays with Jatoba extracts can be justified based on several factors, particularly in early-stage or broad-spectrum studies:
1) Vero cells are widely used in toxicology due to their stable and predictable growth characteristics, non-tumorigenic nature, and well-documented response to various compounds. They offer reliable, consistent results, which can be critical when screening novel compounds like Jatoba extracts.
2) The literature on Vero cells in toxicity assays provides a robust baseline for interpreting results, especially when testing plant extracts with potentially unknown bioactivity;
3) As a non-tumor cell line, Vero cells avoid the complexities that arise with cancer cells, such as altered metabolism and proliferation pathways. Their alignment with non-transformed human cells, though not perfectly human-specific, makes the methodology as close to human-specific as possible, allowing for an effective baseline.
4) Vero cells, being relatively cost-effective and technically simpler to maintain compared to some primary human cell lines or certain specialized human cell lines, reassure researchers about the practicality of the methodology. This practicality can be advantageous in early toxicity screening, allowing researchers to focus on assessing broad cytotoxic potential without the added complexities of using primary human cells.
5) Since kidneys are a primary route for excreting toxic compounds, using kidney-origin cells (even though they are from monkeys) provides insights into cellular responses relevant to renal processing and toxicity.
Comments 4: Please, improve conclusion section.
Answer 4: Thank you for the careful review of our work. A section dedicated to the specific application prospects of the investigated extracts has been added to the Conclusion section (lines 462-467).